# Glutathione and a Pool of Metabolites Partly Related to Oxidative Stress Are Associated with Low and High Myopia in an Altered Bioenergetic Environment

**DOI:** 10.3390/antiox13050539

**Published:** 2024-04-27

**Authors:** Salvador Mérida, Amparo Návea, Carmen Desco, Bernardo Celda, Mercedes Pardo-Tendero, José Manuel Morales-Tatay, Francisco Bosch-Morell

**Affiliations:** 1Department of Biomedical Sciences, Faculty of Health Sciences, Universidad Cardenal Herrera-CEU, CEU Universities, Alfara del Patriarca, 46115 Valencia, Spain; salvador.merida@uchceu.es (S.M.); carmen.desco@uchceu.es (C.D.); 2Instituto de la Retina y Enfermedades Oculares, 46005 Valencia, Spain; doctoranavea@retinavalencia.com; 3FOM, Fundación de Oftalmología Médica de la Comunidad Valenciana, 46015 Valencia, Spain; 4Physical Chemistry Department, University of Valencia, 46100 Valencia, Spain; bernardo.celda@uv.es; 5Department of Pathology, Medicine and Odontology Faculty, University of Valencia, 46010 Valencia, Spain; m.mercedes.pardo@uv.es; 6INCLIVA Biomedical Research Institute, 46010 Valencia, Spain

**Keywords:** myopia, oxidative stress, metabolomics, glutathione, 2-aminobutyrate, choline, acetate, lactate, glycine, sn-glycero-3-phosphocholine, 2-hydroxybutyrate

## Abstract

Oxidative stress forms part of the molecular basis contributing to the development and manifestation of myopia, a refractive error with associated pathology that is increasingly prevalent worldwide and that subsequently leads to an upsurge in degenerative visual impairment due to conditions that are especially associated with high myopia. The purpose of our study was to examine the interrelation of potential oxidative-stress-related metabolites found in the aqueous humor of high-myopic, low-myopic, and non-myopic patients within a clinical study. We conducted a cross-sectional study, selecting two sets of patients undergoing cataract surgery. The first set, which was used to analyze metabolites through an NMR assay, comprised 116 patients. A total of 59 metabolites were assigned and quantified. The PLS-DA score plot clearly showed a separation with minimal overlap between the HM and control samples. The PLS-DA model allowed us to determine 31 major metabolite differences in the aqueous humor of the study groups. Complementary statistical analysis of the data allowed us to determine six metabolites that presented significant differences among the experimental groups (*p* < 005). A significant number of these metabolites were discovered to have a direct or indirect connection to oxidative stress linked with conditions of myopic eyes. Notably, we identified metabolites associated with bioenergetic pathways and metabolites that have undergone methylation, along with choline and its derivatives. The second set consisted of 73 patients who underwent a glutathione assay. Here, we showed significant variations in both reduced and oxidized glutathione in aqueous humor among all patient groups (*p* < 0.01) for the first time. Axial length, refractive status, and complete ophthalmologic examination were also recorded, and interrelations among metabolic and clinical parameters were evaluated.

## 1. Introduction

Myopia, also known as short-sightedness or near-sightedness, affects one-quarter of the world’s population, significantly impacting visual function. This condition is becoming increasingly prevalent worldwide, particularly in East Asia [1]. Based on current trends, it is estimated that myopia will affect 49.8% of the population by 2050 [2,3]. In patients with myopia, images are focused in front of the retina due to excessive curvature of the cornea or lens and elongation of the eye [4]. As such, myopia is one of the most common forms of refractive error. In some types of myopia, excessive axial elongation of the eyeball leads to thinning of the choroid, retinal pigment epithelium, and scleral layer, particularly at the posterior pole [4,5]. Myopia can be caused by both genetic and environmental factors [1,6,7].

High myopia (HM) is usually considered a refractive error with a cutoff in the range of ≥−5.0 D to −10.0 D (spherical equivalent) concomitant with an excessive elongation of axial length longer than 26.0 mm [8,9,10]. HM is also frequently called pathologic myopia. This condition can lead to degenerative changes in the posterior segment of the eye. These changes can include posterior staphyloma, retina choroidal atrophy, myopic maculopathy, or retinal detachment, leading to blindness in several cases [11].

In a healthy eye, the ciliary processes secrete aqueous humor (AH) into the posterior chamber; AH is a clear fluid found in both the anterior and posterior chambers of the eye. It then flows into the anterior chamber and drains out through the trabecular meshwork and Schlemm’s canal. Neurotransmitters are key mediators for regulating the AH dynamics. AH provides nutrition, removes metabolic waste products, and generally supports the avascular ocular structures [12]. Previously, our research team applied metabolomics for the first time in characterizing human AH [13], finding significant metabolic differences in patients with high and low myopia (CE–MS and LC–MS platform methods). Similarly, a member of our team has been a pioneer in the metabolomic characterization of rat AH using NMR for glaucoma research [14].

Metabolomics, which represent the global profiling of metabolites in biological fluids, cells, and tissues, have the potential to discover new biomarkers for diseases [15]. Hence, through the integration of metabolomics with advanced statistical methodologies, one can discern comprehensive alterations in networks and modified biochemical pathways resulting from a pathological process, even during the initial stages of the disease.

Nuclear magnetic resonance (NMR) spectroscopy is one of the technological options for analyzing metabolic profiles and providing a representation of the phenotype at the molecular level [16]. This technique holds significant advantages over other analytical methods, including reproducibility and simplicity in sample preparation. Conversely, its constraints include low sensitivity and a relatively large volume of the sample (~500 μL). However, the present study illustrates that these limitations can be overcome through the utilization of microprobes (~20 μL).

Oxidative stress forms part of the molecular bases that participate in the growth and appearance of diseases associated with myopia [17]. In previous studies, we found that oxidative stress parameters, total antioxidant capacity, and total nitrite/nitrates are significantly altered in the AH of HM patients [18]. Therefore, in this study, we aimed to build on our previous metabolomic research focusing specifically on the study of oxidative stress in myopia by interrelating potential oxidative-stress-related metabolites found in AH and conducting a clinical study of myopic and non-myopic patients.

## 2. Materials and Methods

### 2.1. Experimental Groups and Patient Selection

We conducted a cross-sectional study, selecting two sets of patients undergoing cataract surgery at the FISABIO Medical Ophthalmological Clinic in Valencia, Spain. All participants provided written informed consent to participate in the study, which was conducted according to the principles of the Declaration of Helsinki. The study received ethical approval from the FISABIO Medical Ophthalmological Clinic Ethics Committee (3001/14).

The first set, comprising 116 eyes from different patients, was used to analyze metabolites in the AH of both myopic and control eyes. Similarly, the second set, consisting of 73 eyes from different patients, was used to measure the concentrations of glutathione (GSH) and the oxidized disulfide glutathione form (GSSG) in the AH of both myopic and control patients. The inclusion criteria were patients with cataracts ready to be operated on, refractive myopia, and hypermetropic defects and astigmatism below +0.75 D and −1.75 D, respectively. The exclusion criteria were concomitant eye disease that could interfere with the results (ongoing maculopathy other than myopic maculopathy, uncontrolled glaucoma with two topical drugs, any stage of uveitis, venous or retinal arterial occlusions, diabetic retinopathy, hypermetropia over +2 D, and astigmatism over −2 D). Patients with uncontrolled high blood pressure despite treatment, diabetes, and a high cholesterol index despite treatment were also excluded. All participants in this study were identified as Caucasian. They presented nuclear and/or cortical cataracts, as well as posterior subcapsular cataracts 2–3, according to the Lens Opacities Classification System III. 

Experimental groups were formed based on the axial length of the eyeball. Eyes with an axial length exceeding 26 mm were classified as HM. Those with axial lengths between 23.5 and 25.9 mm and presenting physiological myopia results from failure of correlation between the refractive components of the eye were classified as low myopia (LM). Eyes with axial lengths below 23.4 mm were used as the control group [18,19,20]. As all the patients had cataracts, refractive status was not used to classify our patients in order to avoid any bias caused by index myopia [21]. Previous works have shown that axial length is not associated with nuclear cataracts in myopic patients [22].

### 2.2. Clinical Exploration

This study received approval from the FISABIO Ethics and Clinical Research Committee and was conducted in full compliance with the Declaration of Helsinki. All patients provided informed consent and underwent a comprehensive ophthalmological examination, including ETDRS best-corrected visual acuity (BCVA), exploration of the anterior segment with a slit-lamp, binocular ophthalmoscopy and wide-field retinography using Optos^®^ Optomap^®^ P200Tx (Optos PLC, Dunfermline, UK) to study fundus oculi, detection of the presence of staphyloma or lack thereof, and classification of maculopathy according to the IMI’s classification [23]. The axial length measurements were taken via interferometry (Zeiss IOLMaster 700^®^, Carl Zeiss Meditec AG, Jena, Germany), and optical coherence tomography (swept-source optic coherence tomography SSOCT TOPCON, Tokyo, Japan) was used to obtain the subfoveal choroidal thickness measurement. This measurement was manually taken with a measuring instrument (caliper) facilitated by the software (9.30.003.02) that came with the apparatus. Measurements were performed by two different observers (C.D.E. and R.A.M.) and were masked for the axial length of the patient. Sections where foveal depression was well observed were selected; a measurement was made for each patient and observer (two measurements per eye in total) at the point corresponding to the center of the fovea. The concordance rate between them was 95%.

### 2.3. Aqueous Humor Sample Recollection

AH samples were collected during cataract surgery. All procedures were performed according to standard protocols. All patients received preoperative therapy, which included antibiotic prophylaxis (Oftalmowell, UCB Pharma, an eyedrop solution containing a combination of gramicidin, neomycin, and polymyxin B) and local anesthesia (Colircusí, Alcon Healthcare, an eyedrop solution containing tetracaine and oxybuprocaine). The eyelids and eyelashes were sterilized, and 5% iodine povidone was instilled on the conjunctival sac. A sterile adhesive dressing was then applied to separate the lashes. The eye surface was rinsed with saline solution, and paracentesis was performed at the site of the planned surgical incision using a sterile, single-use 30 G needle. This allowed for the aspiration of a 120 μL AH sample into a 1 cc sterile single-use syringe. Following this, the surgery proceeded using the standard technique. The collected sample was placed inside an Eppendorf tube, frozen in liquid nitrogen, and stored at −80 °C until it was needed. 

### 2.4. NMR Assay

For NMR measurements, 20 μL of AH was mixed with 2.5 μL of sodium-30-trimethylsilylpropionate-2,2,3,3-d4 (TSP; 0.5 mM) dissolved in deuterium oxide (D2O), and that entire mixture was placed in a 1 mm high-quality NMR tube. The NMR spectra were obtained with a Bruker Avance III DRX 600 (Bruker Biospin GmbH, Rheinstetten, Germany) spectrometer operating at 600.13 MHz equipped with a 1 mm 1H/13C/31P TXI probe. A single pulse with water presaturation was acquired in all samples. The number of transients was 256, and they were collected into 64 k data points with a spectral width of 14 ppm and a recycle delay of 1 s. The nominal temperature of the sample was kept at 310 K. In addition, 2D 1H-1H TOCSY and 2D 1H-13C-HSQC NMR experiments were recorded on some selected samples for the assignment of relevant metabolites. Spectra were manually phase corrected, the baseline was adjusted, and the chemical shift was referenced to the TSP signal using MestReNova 6.2 (Mestrelab Research S.L., Santiago de Compostela, Spain). The metabolite spin systems and resonances were identified and quantified using data obtained from previously reported data [14] and the commercial resonance database Chenomx NMR Suite Profiler (Chenomx NMR Suite 8.1, Chenomx Inc., Edmonton, AB, Canada). The final metabolite levels were calculated in arbitrary units as the area under the peak normalized to the total metabolic concentration.

### 2.5. Glutathione Assay

The second set consisting of 73 eyes from different patients was used to analyze oxidative-stress-related markers, specifically, reduced glutathione (GSH) and its ratio with oxidized glutathione (GSSG). The use of a second set was justified because the technique used required the immediate acidification of the samples. Consequently, the procedure was based on the initial formation of S-carboxymethyl derivatives of free thiols with iodoacetic acid followed by the conversion of free amino groups to 2,4-dinitrophenyl derivatives through a reaction with 1-fluoro-2,4-dinitrobenzene. Glutathione was measured in AH via HPLC (Gilson International B.V., Middleton, WI, USA) in accordance with a widely referenced method [24]. A 3-Spherisorb NH2 5 µm, 250 mm × 4.6 mm column (Waters Cromatography, Milford, MA, USA) was used at a 1 mL/min flow rate. Values are expressed as nmol/mg of protein.

### 2.6. Statistical Analysis

Chemometric analyses were performed with PLS ToolBox 8.0 (Eigenvector Inc., Wenatchee, WA, USA) and using in-house scripts developed by the Metabolomic Laboratory for data analysis in MATLAB (The MathWorks Inc., Natick, MA, USA). The NMR spectra were statistically analyzed by using principal component analysis (PCA) to check the homogeneity of the dataset and to identify and exclude possible outliers. Outliers were defined as samples that were situated outside the 95% confidence interval of the Hoteling T-squared distribution. Partial Least Squares Discriminant Analysis (PLS-DA) is a supervised extension of PCA that is used to distinguish two or more classes by searching for variables that maximize the separation between the groups of samples. The datasets used in this analysis were autoscaled prior to multivariate analysis. The determination of statistical significance between the means of different groups was performed using one-way analysis of variance (ANOVA). The spectral regions responsible for the classification of the models were identified using the variable importance in projections (VIP) coefficients obtained during the PLS-DA. The threshold used for VIP selection was ≥ 1. Spectral regions with high VIP coefficients were more important in terms of providing class separation during the analysis, while those with very small VIP coefficients provided only a small contribution to the classification.

Complementary statistical analyses were conducted using the IBM SPSS software (Version 29.0, IBM Corp., Armonk, NY, USA). The Kolmogorov–Smirnov test was used to verify data normality (*p* > 0.05). If the homogeneity of variances was indicated (*p* ˂ 0.05) by Levene’s test, an ANOVA was performed with Tukey’s test as a post hoc analysis. For variables that did not present homogeneity of variances, the Kruskal–Wallis test was used for non-parametric analysis. Pearson’s or Spearman’s correlations were employed to examine the association between two variables. Statistical significance was set to *p* ˂ 0.05. Values are expressed as the mean ± standard deviation (SD).

## 3. Results

### 3.1. The Clinical Data Recorded Were Very Consistent

In the first set of 116 patients, 40 were controls, 48 had LM, and 28 had HM. Their respective mean ages were 74.7 ± 7.1, 72.8 ± 11.9, and 68.1 ± 13.5 years, with 56% women and 44% men. Key clinical features such as axial length, spherical equivalent, macular thickness, and choroidal thickness are documented in Table 1. The choroidal thickness, as measured via optical coherence tomography, was notably reduced in the HM group (125.5 ± 106.7 μm) compared to the control (237.1 ± 64.5 μm) and LM (196.2 ± 77.3 μm) groups, with a significance of 0.01 (Table 1). Since there were no significant differences between the sexes, the patients were considered as a single group.

To analyze the pathologies developed in our study patients, we followed the IMI’s classification, which includes five degrees of macular degeneration (Table 2). In 93.7% of the control patients, the eyes appeared normal, with tessellation being observed in 6.3% of the cases. Macular degeneration was more predominant in the HM group, along with diffuse chorioretinal atrophy, lacquer cracks, and macular atrophy, which were observed in 85.2% of the cases (Table 2).

Ocular biometric parameters were also characterized through post-operative best-corrected visual acuity (BCVA post) and corneal parameters—keratometry, flat keratometry (K1), and steep keratometry (K2)—all of which are shown in Figure 1. 

Spearman’s rank correlation coefficient of the strength of a linear relationship between paired clinical data (Figure 2, * *p* < 0.05) showed the consistency in axial length, reciprocal length (diopters), and choroidal thickness as relevant parameters in the development of myopia. In this way, many of these correlations between clinical variables reached a significance level of *p* < 0.01, including those between the axial length and reciprocal length (ρ = 0.745, *p* < 0.01), reciprocal length and choroidal thickness (ρ = −0.351, *p* < 0.01), and axial length and choroidal thickness (ρ = −0.427, *p* < 0.01). 

### 3.2. PLS-DA Confirmed the Significant Differences between the Control and High-Myopia Groups

A total of 59 metabolites were assigned and quantified in 40 AH samples from the control subjects, 48 samples for the low-myopia patients, and 28 samples for the HM patients. The list of chemical shifts that were used for metabolite assignment is presented in Appendix A. The metabolic profiles of the three groups were compared by means of PCA, which is an unsupervised test for the homogeneity of the set of samples. Based on the PCA, two samples were identified as outliers and excluded from further statistical analyses. Outliers were defined as samples located far outside the 95% confidence interval on the score scatter plot. The exploratory analysis did not show a spontaneous grouping of the samples according to the classification into control and myopia patients, so PLS-DA was used to maximize the separation of the three groups and to reveal specific metabolic changes between the defined ones. Three independent discriminative models were created.

In a first approach, we constructed a model aimed at separating the samples into the three previously defined groups. The outcome is shown in Figure 3A, where we observe an initial division of the samples into three groups. However, the overlap among them is so significant that it lacks discriminatory potential as evidenced by the ROC curves constructed for each of the three groups (cross-validated ROC curve close to the diagonal line), along with the correspondingly low values of the area under the curve obtained (Appendix A). In the second stage, we constructed a new supervised classification model with the aim of separating the LM samples from the control samples (Appendix A). Unlike in the previous model, in this case, the overlap zone between the two groups was more substantial, implying that the LM patients exhibited a metabolomic profile more similar to that of the controls. The ROC curve (Appendix A) confirmed that the model exhibited a very limited classification capacity. Finally, we constructed a new discriminative model designed to maximize the separation between the two most extreme groups—high myopes versus controls (Figure 3B). The PLS-DA score plot clearly showed a separation with minimum overlap between the HM and control samples, confirming the existence of significant differences in the metabolic profiles between the two groups. The area under the ROC curve of the final model (Figure 3C) was 0.74. This model had a Q2 value of 0.77 (a Q2 value superior to 0.5 is generally considered to be a good predictor) and an R2 value of 0.65 (error of calibration). The final cross-validated sensitivity and specificity of the model were 82.5% and 71.4%, respectively. The positive and negative predictive values were 93% and 96%, respectively. 

### 3.3. Main Metabolite Differences in the Aqueous Humor of the Study Groups

The spectral regions and peaks with the highest contribution in the HM vs. CTRL PLS-DA model (VIP scores of > 1) allowed us to determine the major metabolic differences in the AH of the study groups (Figure 3D and Appendix A). 

These included metabolites involved in bioenergetic pathways, such as glucose and lactate, as well as ketone bodies, such as 2-Hydroxybutyrate and 3-Hydroxybutyrate. We also found direct oxidative-stress-related metabolites, including 2-aminobutyrate and glutathione. Additionally, choline and its derivatives (o-acetylcholine, o-phosphocholine, and sn-Glycero-3-phosphocholine) and methylated metabolites were found, among other contributing metabolites.

A complementary statistical analysis of the data (IBM SPSS software, version 29.0) allowed us to determine six metabolites that presented significant differences among the experimental groups (*p* < 0.05). These compounds are listed in Table 3.

Additionally, Spearman’s rank correlation coefficient was used to determine the strength of the linear relationships between all paired AH metabolites (Appendix A, * *p* < 0.05) and between the clinical data and AH metabolites (Appendix A, * *p* < 0.05). Figure 4 highlights the linear relationships (*p* < 0.01) between metabolites that were found to have statistically significant differences, as listed in Table 3, as well as metabolites that contributed to the discrimination between the HM and control groups in our PLS-DA model (Figure 3D, VIP scores > 1.0).

### 3.4. All Derivate Glutathione Biomarkers Showed Significant Differences among All Patient Groups

The 73 patients in the second set were distributed as follows: 33 were the controls, 20 were in the LM group, and 20 were in the HM group. The clinical characteristics of the studied patients in this set showed similar values to those of the first set of patients. The GSH level in the LM group (Figure 5A) was significantly lower than that in the control group (*p* ≤ 0.01). Similarly, we observed a significant decrease in the reduced glutathione (GSH) concentration in the HM group versus the control and LM groups. Concomitantly, the level of oxidized glutathione was significantly higher in the study groups than in the control group (Figure 5B, *p* < 0.01). Therefore, a significant increase in the GSSG concentration in the HM group versus both the control and LM groups was also observed. Figure 5C shows the GSH/GSSG ratio to highlight these results.

## 4. Discussion

### 4.1. NMR Metabolomic and Chemometric Approaches

Metabolites are the endpoints of many biological and physiological processes, and metabolomics may, thereby, provide snapshots of different physiological states. The use of 1H NMR spectroscopy allowed us to identify and quantify sets of metabolites in human AH using a simple sample preparation method. The use of a 1 mm microprobe provided us with the additional advantage of being able to use the smallest amount of sample to run all high-resolution NMR experiments with outstanding sensitivity, without sacrificing the quality of the spectroscopic signal. Additionally, the metabolite profile could be acquired relatively rapidly (10 min with a short routine), with a sufficient ability to evaluate subtle differences even in the early stages of the disease. By employing the combination of NMR metabolomic and chemometric approaches, we were able to establish a metabolic snapshot of the different physiological states associated with myopic and control subjects. We analyzed the mean differences, multivariate PCA and PLS-DA models, VIP scores, and relative fold changes between metabolically healthy individuals and those with high and low myopia to gain a better understanding of possible altered metabolic pathways. Furthermore, NMR metabolomics is an effective approach for identifying potential disease biomarkers in human AH.

### 4.2. Direct Oxidative-Stress-Related Metabolites

Recently, in the serum of myopic children and adolescents, Du et al. (2020) found selected pathways where metabolic features showed regulations with oxidative stress, including lysine degradation, arginine and proline metabolism, arachidonic acid metabolism, linoleic acid metabolism, and sphingolipid metabolism [25]. In our study, L-2-aminobutyric acid, a conjugate acid of an L-2-aminobutyrate metabolite, showed significantly higher levels in the HM group. 2-aminobutyric acid is a non-protein amino acid with relevant physiological skills linked to oxidative stress in different systems [26,27], including the human corneal endothelium [28]. This metabolite is a key intermediate in the biosynthesis of ophthalmic acid (γ-glutamyl-L-2-aminobutyryl-glycine), and it is significantly elevated in oxidative stress conditions [27,29]. Since ophthalmic acid, which was originally isolated from a calf eye lens, is a tripeptide analog of glutathione, it has been proposed that 2-aminobutyric acid regulates glutathione homeostasis [25]. Moreover, in a recent rat model study of neuroprotection in glaucoma [30] against mitochondrial and metabolic dysfunction, some low-weight metabolites, such as 2-aminobutyric acid, creatine, and glycerophosphocholine, seemed to be relevant in the retina. Our results seem to confirm that similar methods are active in high-myopia patients. 

Both metabolites (2-aminobutyric acid and glutathione) were relevant in our PLS-DA model. Similarly, concomitant with the increased levels of 2-aminobutyric acid, we found a depletion of GSH levels in the patients of the second set. In fact, ophthalmic acid is formed by the same enzymes as glutathione, with the incorporation of 2-aminobutyric acid instead of Z-cysteine in the first biosynthetic step in such a way that the dismissal of GSH by the oxidative stress environment induced increased 2-aminobutyric acid levels and ophthalmate synthesis [26]. Interestingly, 2-aminobutyric acid was one of the two metabolites that presented the most significant correlations with the clinical parameters of pathological myopia, such as macular thickness, peripapillary atrophy, lacquer tracks, and diffuse chorioretinal atrophy (Appendix A, *p* < 0.05).

Oxidative stress participates in the growth and appearance of diseases associated with myopia [17,18]. In this study, we implemented a new set of patients (second set) to measure reduced glutathione (gamma-glutamyl-cysteinyl-glycine, GSH) and glutathione disulfide (GSSG) levels in samples of AH from myopic patients for first time. In animal cells, GSH is the most abundant low-molecular-weight antioxidant thiol and co-substrate for detoxification enzymes such as glutathione peroxidase. The redox state of the GSH/GSSG couple serves as a leading indicator of a redox environment. Both GSH and GSSG are exported from cells through multidrug resistance protein; extracellular GSH is metabolized by membrane-bound γ-glutamyl transpeptidase into cysteinylglycine and γ-glutamyl products, and dipeptidase hydrolyzes cysteinylglycine to cysteine and glycine [31]. We observed markedly diminished GSH levels in the LM and HM groups in the second set of patients. Simultaneously, the levels of oxidized glutathione (GSSG) were also increased in the LM and HM groups. In the same way, the GSH/GSSG ratio showed marked differences among the groups. All of these results reaffirm the role of oxidative stress in the evolution of myopia as a disease and are in concordance with some of the altered metabolites related to oxidative stress found in the first set, such as 2-aminobutyrate, glycine, and lactate. Interestingly, recent works have postulated the redox communication network as a regulator of metabolism [32].

### 4.3. Scleral Remodeling and Metabolites Involved in Bioenergetic Pathways

There are different pathways connecting the retina to scleral remodeling [33]. It is known that chemical signals from photoreceptors may be transferred to the choroid via the retinal pigment epithelium and, finally, modify the scleral remodeling [33]. On the one hand, the retinal pigment epithelium presents receptors for key signaling molecules, such as cholinergic, glycine, and dopamine receptors. On the other hand, scleral fibroblasts express growth factors, including insulin [34] and glucagon [35], which also participate in scleral remodeling. We found altered levels of several metabolites related to these pathways (Table 3 and Figure 3D), including neurotransmitters responsible for mediating interactions in retinal cells, such as glycine and choline/acetylcholine derivates, as well as glucagon and insulin receptors expressed in scleral fibroblasts. 

Glycine plays significant roles in human metabolism and nutrition. Therefore, glycine is present in macromolecules such as collagen, provides flexibility of active sites on enzymes, and plays crucial roles as a neurotransmitter. In our study, the glycine levels significantly decreased in patients with HM. There are several ways in which glycine might participate in the pathology of the myopic eye, presenting a certain protective effect. Firstly, glycine modulates the production of superoxides and the synthesis of cytokines by modifying the intracellular Ca^2+^ levels [36]. Glycine is also released by AII amacrine cells to inhibit OFF-cone bipolar cell neurotransmission [37]. Interestingly, AII amacrine cells show increased phosphorylation of Cx36, which could indicate increased functional gap junction coupling in the myopic retina [38]. Glycine is likewise one of the three amino acids that constitute ophthalmic acid, the tripeptide analog of GSH [26]. Moreover, recent studies showed that genetic variations in glycine receptor alpha 2 expression may be considered HM-causing genes by harming photoperception and visual transmission [39].

Glucose and lactate play known crucial roles in our body’s metabolic processes. We found a positive correlation of glucose with the axial length of the studied patients (Appendix A, ρ = 0.226, *p* < 0.05) and a lightly negative correlation for lactate. Glucose levels contributed to the discrimination between groups in our PLS-DA model (Figure 3D, VIP scores > 1.0, HM vs. control group; Appendix A, VIP scores > 1.0, LM vs. control). Li et al. [40] recently evaluated multiple glycemic traits related to the risk of myopia, finding that low adiponectin levels and high hemoglobin A1c were linked to an increased risk of myopia. In our study, we also found that methanol levels contributed to the discrimination between groups in our PLS-DA model (Figure 3D, VIP scores > 1.0, HM vs. control group; Appendix A, VIP scores > 1.0, LM vs. control). Zhu [41] proposed that an increase in cellular glucose levels would lead to increased metabolic formation of endogenous methanol, which may be further metabolically converted into formic acid or formate, a cytotoxic metabolite that we also found in the AH of patients (Appendix A). In this way, the methanol/formic acid biotransformation might cause cytotoxicity by directly inhibiting the cytochrome oxidase complex of the mitochondrial respiratory chain, thereby reducing the cellular ATP production, increasing ROS production, and reducing NAD to NADH.

Lactate is a three-carbon molecule generated through glycolytic metabolism that acts as a relevant carbohydrate fuel and is especially relevant in a fasted state [42]. In our study, the LM group showed a little and non-significant increase in lactate levels. However, we found that the HM group displayed significantly diminished levels of lactate. Arising evidence connects the pyruvate dehydrogenase kinase–lactic acid axis and the pathophysiology of neurological disorders [43]. Traditionally, hypoxia linked to ischemic injury directly leads to lactate accumulation. Therefore, Lin et al. [44] recently proposed that, in an animal model, scleral glycolysis contributed to myopia by promoting fibroblast-to-myofibroblast transdifferentiation via lactate-induced histone lactylation. Our results agree in the LM group but not in the HM group. This paradoxical result may be related to the fact that lactate is also admitted as a signaling molecule involved in cell survival, as a potential energy substrate, and as a redox buffer molecule that equilibrates the NADH/NAD ratio [45]. Recent evidence suggested that lactate treatment lessened brain damage and improved behavior in neonatal rat models of hypoxic ischemia [46], indicating that lactate may play a neuroprotective role in ischemic conditions. Thus, elevated levels of lactate may be related to the development of myopia, and diminishing levels of lactate might be linked to advanced damage in the highly myopic eye in different ways. On the one hand, lactate diminishes oxidative-stress-induced cell death and promotes oxidative stress resistance via the UPR and p38MAPK pathways [45], and on the other hand, lactate triggers an immunomodulatory response via G-protein-coupled receptor 81 and reduces the production of pro-inflammatory cytokines [47]. Likewise, lactate interferes in classical microglial polarization, decreasing neuroinflammatory parameters [48].

Ketone bodies, 2-hydroxybutyrate, and 3-hydroxybutyrate metabolites highly contributed to the PLS-DA model for discriminating among groups, reaching VIP scores above 2.0 (Figure 3D and Appendix A). These two metabolites are derived from fatty acid oxidation and serve as a fuel source for peripheral tissues, including the brain and eye [49,50]. In the retina, 3-hydroxybutyrate, acetoacetate, and acetone serve as energy sources during nutritional deprivation [50]. In addition, 2-hydroxybutyrate works as an energy substrate for maintaining metabolic homeostasis and organ integrity in response to ischemia and reperfusion by suppressing lipolysis, mitochondrial dysfunction, and oxidative stress [51,52].

Creatine phosphate was the other metabolite that presented the most significant correlations with the clinical parameters of pathological myopia, correlating significantly with choroidal thickness, peripapillary atrophy, lacquer tracks, and diffuse chorioretinal atrophy (Appendix A, *p* < 0.05). Creatine phosphate is a phosphorylated form of creatine that serves as a rapidly mobilizable reserve of high-energy phosphates. Both creatine and phosphocreatine are required to preserve ATP, which is needed for normal retinal function and development [53]. In our study, creatine was one of the relevant metabolites in our PLS-DA model. It has been proposed that phosphocreatine may reduce the ADP-induced stimulation of mitochondrial metabolism, and it would be related to the inability of the neuroprotective role of creatine to protect retinal neurons in vivo [54]. In any case, these correlations reaffirm the great relevance of oxidative stress in pathological myopia.

In a previous metabolomic study using serum samples from myopic patients, Ke et al. (2021) discovered intermediates in energy metabolism, such as citric acid and oxaloacetic acid [55]. It has been proposed that a surge in energy metabolism could lead to a rise in extracellular adenosine levels [56]. This, in turn, stimulates the activation of adenosine receptors, which contribute to the control of eye growth [57]. Interestingly, adenosine has been suggested to exert a consistent inhibitory impact on the release of acetylcholine in the retina by interacting with adenosine 1 receptors, as in other cholinergic synapses [58].

### 4.4. Choline and Choline Derivates

Choline and choline derivates, such as o-acetylcholine, o-phosphocholine, and sn-glycero-3-phosphocholine, showed relevant variations in their levels in the AH of the study groups (Figure 3D, VIP scores > 1, PLS-DA model; Appendix A, VIP scores > 1, PLS-DA model). We found elevated levels of choline in the AH samples from both low- and high-myopic eyes. Choline has been suggested to have key roles in neurotransmission, the structural integrity of membranes, and methyl group metabolism, and it is a precursor for the synthesis of acetylcholine, a relevant neurotransmitter not only in muscle contraction signaling but also in the brain and retina [59,60]. In fact, acetylcholine plays an important role in the developing retina and regulates the growth of the eye [61]. We also found higher levels of acetate in myopic samples (Figure 3D, VIP scores > 1, PLS-DA model; Appendix A, VIP scores > 1, PLS-DA model)—specifically, high acetate levels were significant in the LM group. As is known, acetylcholinesterase immediately hydrolyzes acetylcholine into acetate and choline, terminating neuronal transmission and signaling between synapses. Interestingly, choline and acetate levels also showed a significant and positive correlation (Appendix A, ρ = 0.183, *p* < 0.05).

We did not find significant differences in acetylcholine levels between groups. However, acetylcholine played a role in distinguishing between the HM and control groups in our PLS-DA model. It showed significant correlations with several pertinent clinical variables, such as axial length (Appendix A, ρ = −0.194, *p* < 0.05), diopters (Appendix A, ρ = −0.276, *p* < 0.01), and the spherical equivalent (Appendix A, ρ = 0.214, *p* < 0.05). At the same time, it demonstrated a significant and positive correlation with choline, phosphocholine, and sn-glycero-3-phosphocholine. In the eye, the five subtypes of muscarinic acetylcholine receptors are broadly distributed and exert many functions, such as neurotransmission, modulation of the intraocular pressure, tear secretion, regulation of vascular diameter in the retina, and contraction of the pupillary constrictor muscle, together with the ciliary muscle, which regulates accommodation [59]. The M1 receptor could be essential for the survival of retinal ganglion cells and helps to increase nitric oxide production from neuronal nitric oxide synthase [62]. The M2 receptor is relevant in the development of myopia by delaying scleral fibroblast cell proliferation and additionally reducing scleral remodeling [63], and the M3 receptors intervene in the cholinergic response in the retinal arterioles and the ophthalmic artery [64]. Conversely, nicotinic acetylcholine receptors (nAChR) have been shown to be present in both the retina and choroid [65]. Endogenous activation of nAChR contributes to the increased incidence of choroidal neovascularization. Moreover, nicotinic acetylcholine receptors (α3, α4, α6, α7, β2, and β4 subunits) in amacrine and ganglion cells have been observed in the retinas of Rhesus monkeys and rabbits [59]. These retinal nicotinic acetylcholine receptors are involved in the processing of visual information and could also influence the progression of retina pathologies, as well as the formation of new blood vessels in the eye [59].

Another choline derivative, sn-glycero-3-phosphocholine, showed diminished levels in myopic samples, and this decrease was significant in the LM group. sn-Glycero-3-phosphocholine plays a role as an osmoprotectant by stabilizing intracellular macromolecules and by protecting inner medullary cells [66]. Interestingly, in recent research, Wu et al. (2023) discovered that the degree of myopia was significantly influenced by the corneal levels of various metabolites, including derivatives of sn-glycero-3-phosphocholine [67]. We did not find further similarities to this previous study in our work. Traditionally, a thinner choroid has been related to the presence of osmotically active molecules [68]. The blood-ocular barriers, including the blood-aqueous barrier and the blood-retinal barrier, are essential for normal visual function and maintaining the eye as a privileged site. While these barriers can be affected by various conditions that affect the eye or have a marked effect on the blood composition [69], the specific alteration of these barriers in axial myopia is not well-established. Therefore, sn-glycero-3-phosphocholine is a precursor of choline mediated by glycerophosphodiester phosphodiesterases, which seems to be in concordance with the high levels of choline found. In fact, the lipid composition of the retina is mostly represented by glycerophospholipids—remarkably, sn-glycero-3-phosphocholine and glycerophosphoethanolamine [70]. Thus, diminished levels of sn-glycero-3-phosphocholine may indicate membrane cell damage, osmodysregulation, and alterations in the biosynthesis of choline in the retina.

### 4.5. Methylated Metabolites

Methylated metabolites, such as τ-methylhistidine, π-methylhistidine, dimethylamine, 1,7-dimethylxanthine, trimethylamine, and N-nitrosodimethylamine (Figure 3D and Appendix A, VIP scores > 1, PLS-DA model), contributed to the discrimination between groups in our PLS-DA model. In a previous work [13], our research team found various metabolites related to methylation processes. This finding has been confirmed by other researchers in retinal eye diseases related to oxidative stress, such as glaucoma, dry eye syndrome, and diabetic retinopathy [71].

The formation of these methyl metabolites relies on methyltransferases, which transfer methyl groups from S-adenosylmethionine (SAM). Wang et al. [72] proposed that the consumption of methyl groups to generate these methylated metabolites might limit the availability of SAM for DNA methylation in the aging eye. DNA methylation is generally thought to be a gene-repressing modification. Moreover, many of these enzymes responsible for preserving the methylated status of DNA are redox-sensitive [71]. Therefore, the regulation of oxidative stress increased DNA methyltransferase 1 binding in retinal endothelial cells [72]. In a recent study focused on DNA methylation in age-related macular degeneration, Advani et al. [73] revealed significant epigenetic regulation of the immune response and metabolism, including the glutathione pathway and glycolysis.

All of these findings suggest that both DNA methylation and oxidative stress play significant roles in myopia. However, the exact mechanisms and their interplay are complex and require further research.

### 4.6. Study Limitations

The age of the patients was high, since the samples were taken from patients undergoing cataract processes. Nevertheless, the groups had similar average ages, and there were no significant differences between them. In fact, the group with HM had the lowest non-significant average age.

The hypothesis that there is a trend of increasing axial myopia with age-related cataracts is still under discussion. Pan et al. (2013) demonstrated that myopia—but not axial length—may be associated with nuclear cataracts [22]. Regardless, the clinical data recorded in our study are very consistent, reaching a significance level of *p* < 0.01 between the main clinical parameters according to the Spearman correlation.

## 5. Conclusions

In the first set of patients, we identified and quantified a total of 59 metabolites. The PLS-DA score plot clearly demonstrated a separation with minimal overlap between HM and control samples. This model enabled us to identify 31 major metabolite differences in the AH of the study groups. Many of these metabolites have been found to be directly or indirectly related to the oxidative stress associated with myopic eye conditions. Interestingly, we found metabolites involved in bioenergetic pathways, methylated metabolites, and choline and choline derivatives. A complementary statistical analysis of the data allowed us to identify six metabolites that showed significant differences among the experimental groups. Additionally, in a second set of patients, we were able to demonstrate significant variations in both reduced and oxidized glutathione in the AH across all patient groups for the first time.

## Figures and Tables

**Figure 1 antioxidants-13-00539-f001:**
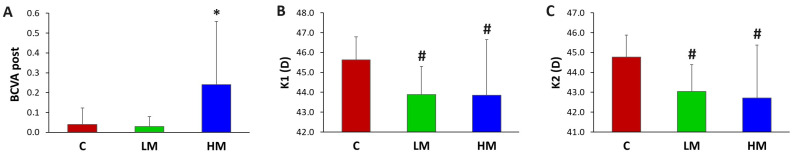
Post-operative best-corrected visual acuity (**A**) and corneal parameters K1 (**B**) and K2 (**C**). Each value is shown as the mean ± SE. * *p* ≤ 0.01 vs. the control and LM groups. # *p* ≤ 0.01 vs. the control group. D: diopters.

**Figure 2 antioxidants-13-00539-f002:**
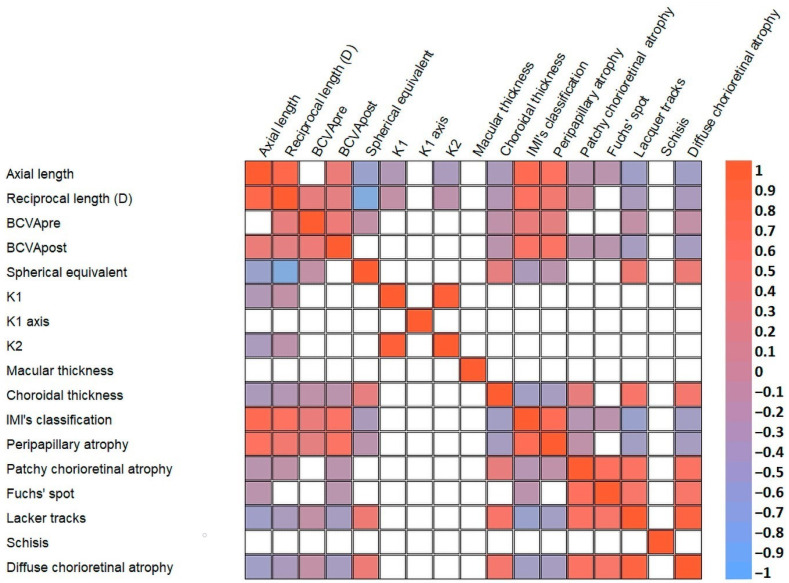
Spearman’s rank correlation coefficient of the strength of the linear relationships between pairs of clinical data. Tabular data are presented as heatmaps (*p* < 0.05): blue (negative correlation) and red (positive correlation). D: diopters.

**Figure 3 antioxidants-13-00539-f003:**
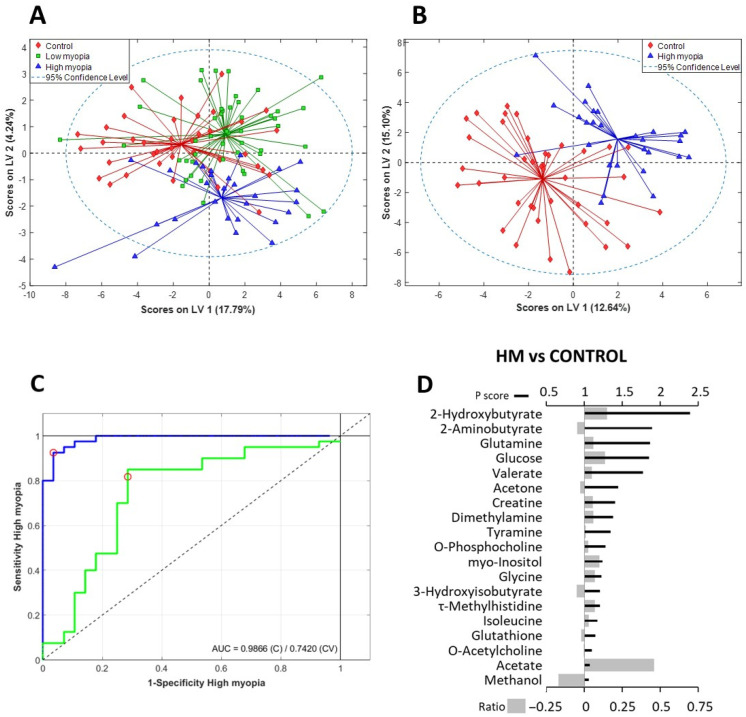
(**A**) Scores plot of the PLS-DA of the metabolome for the discrimination between the control group (red diamonds), LM group (green squares), and HM group (blue triangles). (**B**) Score plot of the PLS-DA for the discrimination between the control group (red diamonds) and HM group (blue triangles). All models were built using the first two latent variables (LVs). (**C**) Receiver-operating curve (ROC) analysis for discrimination between the HM group and control samples showing the prediction capacity of the model. (**D**) VIP score and relative fold change bar plot of the HM/control discriminative model (blue line: estimated ROC curve; green line: cross-validated ROC curve; red circle: model threshold). VIP scores are represented as thick black lines (scale on the top), and the relative fold change was calculated as follows: (Metabolite mean concentration of HM—metabolite mean concentration control)/Metabolite mean concentration of control. These are represented by gray bars (scale on the bottom).

**Figure 4 antioxidants-13-00539-f004:**
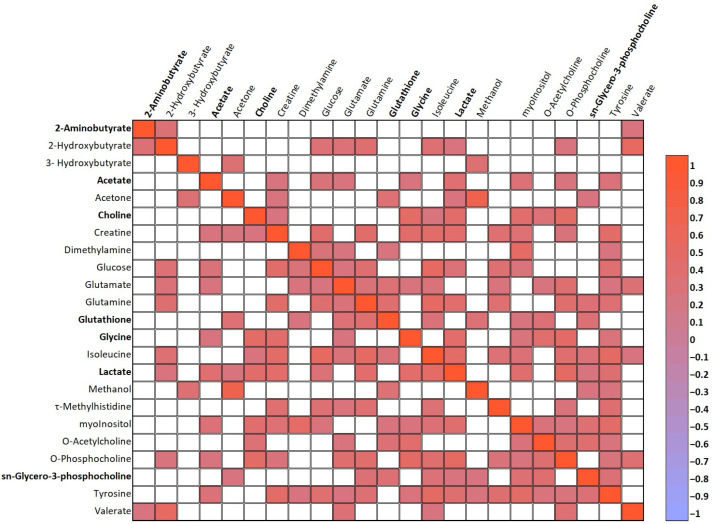
Spearman’s rank correlation coefficient of the strength of the linear relationships between the metabolites present in AH and found to be statistically significant according to the differences among experimental groups and the metabolites that contributed to the discrimination between the HM and control groups in our PLS-DA model. Tabular data are presented as a heatmap (*p* < 0.01): blue (negative correlation) and red (positive correlation). All the metabolites that showed significant differences among groups are highlighted in bold.

**Figure 5 antioxidants-13-00539-f005:**
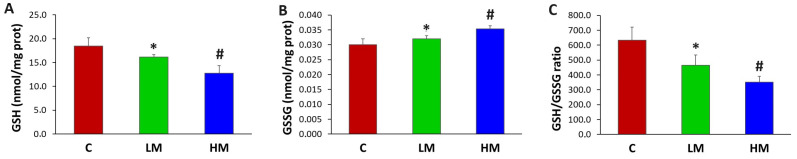
(**A**) Reduced glutathione (GSH), (**B**) oxidized glutathione (GSSG), and (**C**) the GSH/GSSG ratio. Each value is shown as the mean ± SE. * *p* ≤ 0.01 vs. the control group. # *p* ≤ 0.01 vs. the control and LM groups.

**Table 1 antioxidants-13-00539-t001:** Relevant clinical characteristics of the studied patients. Groups were distributed as the HM, LM, and control groups for emmetropia according to their axial length. Each value is shown as the mean ± SE.

	N	Axial Length (mm)	Spherical Equivalent	Macular Thickness (μm)	Choroidal Thickness (μm)
C	40	22.6 ± 0.4	−1.1 ± 1.5	259.2 ± 33.0	237.1 ± 64.5
LM	48	24.2 ± 0.7 *	−3.4 ± 3.0 *	249.7 ± 26.3	196.2 ± 77.3
HM	28	28.4 ± 1.9 *#	−9.7 ± 7.6 *#	269.7 ± 42.4	125.5 ± 106.7 *#

* *p* ≤ 0.01 vs. the control group, and # *p* ≤ 0.01 vs. the low-myopia group.

**Table 2 antioxidants-13-00539-t002:** Distribution of maculopathies in the eyes of the studied patients following the IMI’s classification expressed as the percentage of the entire study population.

Degrees of Maculopathy	C	LM	HM
No myopic degenerative retinal lesion	29.5	32.8	0.8
Tessellated fundus	1.6	5.7	3.3
Diffuse chorioretinal atrophy	0	0	5.7
Patchy chorioretinal atrophy	0.2	0.5	1.3
Macular atrophy	0	0	5.7
PLUS: NEOVASCULARIZATION/FUCHS	0	0	4.1
PLUS: LACQUER CRACKS	0	0	4.9

**Table 3 antioxidants-13-00539-t003:** A list of compounds showing significant differences among the experimental groups.

ID	Formula	LM %change	HM %change
2-Aminobutyrate	C_4_H_9_NO_2_	−12.54	+18.64 #
Acetate	C_2_H_4_O_2_	+40.87 *	+12.72
Choline	C_5_H_14_NO	+119.66 *	+9.39 *#
Glycine	C_2_H_5_NO_2_	+5.13	−21.45 #
Lactate	C_3_H_6_O_3_	+2.22	−8.40 #
sn-Glycero-3-phosphocholine	C_8_H_21_NO_6_P	−19.19 *	−15.47

The percentage change value was calculated with respect to the control group. * *p* < 0.05 vs. the control group. # *p* < 0.05 vs. the LM group.

## Data Availability

The data presented in this study are available in this article.

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
