# Peer review of "Glutathione and a Pool of Metabolites Partly Related to Oxidative Stress Are Associated with Low and High Myopia in an Altered Bioenergetic Environment"

_antioxidants, 2024, doi:10.3390/antiox13050539_

Round 1
Reviewer 1 Report
The paper by dr. Mérida and colleagues reports a detailed metabolomic analysis of the vitreous humor of patients displaying varying degrees of myopia. The results report important differences in metabolites involved in oxidative stress and in metabolites involved in bioenergetic pathways, methylated metabolites, and choline and choline-related molecules. In addition, differences in reduced and oxidized glutathione have also been reported. Overall, this is a very “dense” work performed with technical skill and appropriate statistical analyses.
I only have a few observations that the Authors may want to consider:
Line 49: abbreviation HM is defined here, but it is not always used in place of high myopia, as it should be. In addition, this abbreviation is re-defined in other places (lines 108 and 202). Please revise the abbreviations. In addition, high myopia is briefly described, but no information is given on low myopia. I think a short description would help the non-expert readers.
Line 59-60, “… the AH … transports neurotransmitters”: This is not clear. Please explain.
Lilne 80: please explain TAC.
Line 82: please replace specically with specifically
Lines 96 and 169: It is said that glutathione was measured in eye homogenates. What are these homogenates? Usually, a tissue homogenate is obtained subjecting the tissue to mechanical rupture and chemical aggression to dissociate the cells and to break the cell membranes to obtain all the tissue constituents in a homogeneous material (homogenate) from which one can extract proteins, nucleic acids, etc. I don’t think this is the case here. Were the samples for glutathione assay obtained from the AH of the patients? Then these are not eye homogenates.
Table 3: Please revise the explanation of the significances.
Line 396-397, “… such as glucose”: The sentence is a little confused, but it seems that glucose is considered as a metabolite related to glucagon and insulin receptors. Of course, insulin and glucagon acting at their receptors regulate glucose levels in he blood, but this definition sounds a bit strange (at least to me).
Line 405: It should be AII amacrine cells and not “type II”. In addition, the fact that glycine released by AII amacrine cells inhibits OFF cone bipolar cells does not seem to have a direct implication with myopia. Instead, you may want to check the paper at doi: 10.3389/fncel.2020.00124. eCollection 2020, where a modulation of AII amacrine cell coupling has been reported in retinas of myopic eyes.
Line 479-480: acetylcholine is an important neurotransmitter in brain and retina and its actions are not limited to “nervous vessels”. A role of acetylcholine in myopia is discussed in the review of your reference [53]. Probably, some attention to this point would be more appropriate than just a mention of putative roles of acetylcholine in vessels of the central nervous system. In addition, how do you explain the observation that choline increases in HM while acetylcholine does not show any change?
Line 498-503: Muscarinic receptors in the eye are discussed here, but what about nicotinic receptors?
Discussion: the reference to figures and tables that have been already described in the results should be avoided in the discussion.
Author Response
Subject: Revision and resubmission of manuscript 2964807
Dear Reviewer,
Thank you for your guidance, feedback, and corrections, as well as the opportunity to revise our paper titled ‘Glutathione and a pool of metabolites partly related to oxidative stress are associated with low and high myopia in an altered bioenergetic environment.’ The suggestions provided by each reviewer have been immensely helpful. We also appreciate your insightful comments.
We have incorporated the reviewers’ comments and responded to each one individually in the attached file, detailing how we addressed each concern or issue and outlining the changes we have implemented. All authors have approved the revisions.

Reviewer 2 Report
The manuscript ‘Glutathione and a pool of metabolites partly related to oxidative stress are associated with low and high myopia in an altered bioenergetic environment’ provides a descriptive analysis of diverse metabolites in the aqueous humour (AH) of myopic patients. Despite different manuscripts describing oxidative stress-related metabolites in the AH of high myopic patients, the authors of this manuscript include new ones not described previously, provide a comparison with low myopic patients, and justify with different ocular conditions. For this reason, I found the manuscript valuable, but I have some questions/comments that I would appreciate the authors clarifying for me:
· Since patients were classified as LM and HM attending especially to the axial length of the eyeball, I wonder if cataracts may affect the axial length of high myopes. If there is any link, it might be included in the manuscript.
· Also, I wonder if these metabolites may be found in patients with myopic refractive status that comes, for example, from cornea alterations. For this reason, I consider that it might be interesting to make some short mention of the altered permeability of ocular tissues and blood barriers in axial myopia, which perhaps may explain the release of all these metabolites in the aqueous humour or other suggestions by the authors.
· The data in Table 1 regarding choroidal thickness makes me doubt the significant differences between LM and HM since SE seems higher. Could this be checked or justified?
· It would be interesting to compare the metabolites found in their study in the aqueous humour with these or others associated with oxidative stress or bioenergetic environment in other fluids such as tears or plasma in HM patients.
· The age of the patients is high since the sample is taken associated with a cataract process. Could this be a drawback for classifying any of these metabolites as a biomarker of myopia progression?
· Line 50: High myopia cutoff should be ‘≥’-5D to -10D.
· Line 96; 169: I feel that it is inappropriate to talk about ‘eye homogenates’. I suggest changing to aqueous humour.
· Line 102: A parenthesis is missing.
· Figure 1C: Y-axis must be “K2”. It is also recommended to add the units of the flat keratometry in the Y-axis of figure 1B and 1C.

Author Response
Subject: Revision and resubmission of manuscript 2964807
Dear Reviewer,
Thank you for your guidance, feedback, and corrections, as well as the opportunity to revise our paper titled ‘Glutathione and a pool of metabolites partly related to oxidative stress are associated with low and high myopia in an altered bioenergetic environment.’ The suggestions provided by each reviewer have been immensely helpful. We also appreciate your insightful comments.
We have incorporated your comments and responded to each one individually in the attached file, detailing how we addressed each concern or issue and outlining the changes we have implemented. All authors have approved the revisions.

Round 2
Reviewer 1 Report
I have no further comments
The Authors have considered all the observations that I made on the previous version of the manuscript and I am satisfied with their responses.
Reviewer 2 Report
Dear Authors,
The revision of the manuscript ‘Glutathione and a pool of metabolites partly related to oxidative stress are associated with low and high myopia in an altered bioenergetic environment’ justifies perfectly all my previous comments/suggestions and corrects some of the minor mistakes found previously. For this reason, I consider that the manuscript can be published.
Congratulations to all the authors.
Dear Authors,
The revision of the manuscript ‘Glutathione and a pool of metabolites partly related to oxidative stress are associated with low and high myopia in an altered bioenergetic environment’ justifies perfectly all my previous comments/suggestions and corrects some of the minor mistakes found previously. For this reason, I consider that the manuscript can be published.
Congratulations to all the authors.